# Nanofibrous Vildagliptin/PLGA Membranes Accelerate Diabetic Wound Healing by Angiogenesis

**DOI:** 10.3390/ph15111358

**Published:** 2022-11-04

**Authors:** Chen-Hung Lee, Chien-Hao Huang, Kuo-Chun Hung, Shu-Chun Huang, Chi-Ching Kuo, Shih-Jung Liu

**Affiliations:** 1Division of Cardiology, Department of Internal Medicine, Chang Gung Memorial Hospital-Linkou, Chang Gung University College of Medicine, Taoyuan 33302, Taiwan; 2Linkou Medical Center, Division of Hepatology, Department of Gastroenterology and Hepatology, Chang-Gung Memorial Hospital, Taoyuan 33305, Taiwan; 3Department of Physical Medicine and Rehabilitation, Chang Gung Memorial Hospital, New Taipei Municipal Tucheng Hospital, New Taipei 23652, Taiwan; 4Department of Physical Medicine & Rehabilitation, Chang Gung Memorial Hospital-Linkou, Taoyuan 33305, Taiwan; 5College of Medicine, Chang Gung University, Taoyuan 33302, Taiwan; 6Research and Development Center of Smart Textile Technology, Institute of Organic and Polymeric Materials, National Taipei University of Technology, Taipei 10608, Taiwan; 7Bone and Joint Research Center, Department of Orthopedic Surgery, Chang Gung Memorial Hospital-Linkou, Taoyuan 33305, Taiwan; 8Department of Mechanical Engineering, Chang Gung University, Taoyuan 33302, Taiwan

**Keywords:** vildagliptin, nanofibrous membranes, electrospinning, release characteristics, diabetic wound healing

## Abstract

The inhibition of dipeptidyl peptidase-4 (DPP4) significantly enhances the wound closure rate in diabetic patients with chronic foot ulcers. DPP4 inhibitors are only prescribed for enteral, but topical administration, if feasible, to a wound would have more encouraging outcomes. Nanofibrous drug-eluting poly-D-L-lactide-glycolide (PLGA) membranes that sustainably release a high concentration of vildagliptin were prepared to accelerate wound healing in diabetes. Solutions of vildagliptin and PLGA in hexafluoroisopropanol were electrospun into nanofibrous biodegradable membranes. The concentration of the drug released in vitro from the vildagliptin-eluting PLGA membranes was evaluated, and it was found that effective bioactivity of vildagliptin can be discharged from the nanofibrous vildagliptin-eluting membranes for 30 days. Additionally, the electrospun nanofibrous PLGA membranes modified by blending with vildagliptin had smaller fiber diameters (336.0 ± 69.1 nm vs. 743.6 ± 334.3 nm, *p* < 0.001) and pore areas (3405 ± 1437 nm^2^ vs. 8826 ± 4906 nm^2^, *p* < 0.001), as well as a higher hydrophilicity value (95.2 ± 2.2° vs. 113.9 ± 4.9°, *p* = 0.004), and showed a better water-retention ability within 24 h compared with PLGA membranes. The vildagliptin-eluting PLGA membrane also enhanced the diabetic wound closure rate for two weeks (11.4 ± 3.0 vs. 18.7 ± 2.6 %, *p* < 0.001) and the level of the angiogenesis using CD31 expression (1.73 ± 0.39 vs. 0.45 ± 0.17 *p* = 0.006 for Western blot; 2.2 ± 0.5 vs. 0.7 ± 0.1, *p* < 0.001 for immunofluorescence). These results demonstrate that nanofibrous drug-eluting PLGA membranes loaded with vildagliptin are an effective agent for sustained drug release and, therefore, for accelerating cutaneous wound healing in the management of diabetic wounds.

## 1. Introduction

The incidence of lower extremity amputations in patients with diabetes mellitus exceeds 20 times that in the general population, and these amputations account for most of the hospital inpatient visits of such patients. Every half minute, somewhere in the world, part of a limb is amputated as the result of the complication of diabetes [1,2]. The mortality rate due to lower extremity amputation is at least 13% in the first year following the sugary, from 35 to 65% at three years, and up to 80% at five years [3]. Several factors that delay healing of diabetic wounds have been identified by biomedical researchers, including reduced keratinocyte migration, fibroblast proliferation, reduced re-epithelialization, and impaired collagen accumulation in wounds [4,5]. The physiopathology of diabetic wounds is complex and not fully understood; neuropathy, as well as macro- and microangiopathy, delay the healing of diabetic wounds by impairing the response of neo-angiogenesis to ischemia and tissue feeding [6].

Vildagliptin, an incretin hormone glucagon-like peptide-1, may reduce oxidative stress by promoting the vascular production of endothelial growth factor [7] and improving hypoxia-inducible factors-1α upregulation [8]. Vildagliptin is also a dipeptidyl-peptidase inhibitor (DPP-4 inhibitor) that reduces the breakdown of endogenous glucagon-like peptide (GLP)-1 and has the potential to improve metabolic control by increasing insulin secretion. By reducing the need for insulin, the pharmaceutical provides vascular protection and exhibits anti-arteriosclerotic activity and pleotropic effects [9,10]. However, patients usually discontinue their course of treatment owing to the adverse effects of vildagliptin, which include nasopharyngitis, headache, dizziness, back pain, and diarrhea [11,12]. DPP4 inhibitors are only prescribed for enteral, but topical administration, if feasible, to a wound have more encouraging outcomes.

Advances in medical textiles and polymer science have led to the development of several innovative drug delivery systems [13,14]. The topical delivery of vildagliptin via polymer-based nanofibrous membranes in which the drug is directly applied to the skin has the advantage of high selectivity and efficiency. Among biomaterials, poly (lactic-co-glycolic acid) (PLGA) is an approved biocompatible and biodegradable polymer for use in membranes for tissue engineering and as a drug delivery vehicle [15,16,17,18].

Nanofibrous biodegradable vildagliptin-eluting PLGA membranes that are electrospun were developed herein for healing diabetic wounds. Nanofibrous scaffolds are very good candidates for tissue engineering and wound healing as they exhibit adjustable characteristics including degradability, wettability, porosity, good mechanical strengths, and cell adhesive and antimicrobial features and operational handleability [19]. Electrospinning is a fascinating technology for making nanofibrous mats that mimic the structure of the extracellular matrix [20]. The technique includes an electrohydrodynamic procedure, where a liquid droplet is electrified to create a jet, followed by straightening and extension to form fibers [21]. Blending the PLGA membranes with vildagliptin in the dressing of wounds is hypothesized to accelerate angiogenesis and cutaneous wound closure as part of the process of diabetic wound recovery. The diameters and pore areas of electrospun nanofibers were evaluated by scanning electron microscopy (SEM) following electrospinning. The release efficacy of the drugs from vildagliptin-eluting membranes was examined using endothelial progenitor cells (EPCs) migration assay. The effects of the nanofibrous vildagliptin-eluting PLGA membranes on diabetic wound repair were investigated by immunohistochemistry and Western blotting. 

## 2. Result and Discussion

The SEM images of the electrospun nanofibrous membranes (Figure 1) were presented at a magnification of 3000×. The sizes of vildagliptin-eluting PLGA fibers (336.0 ± 69.1 nm) (Figure 1A) were lower than those of the drug-free PLGA nanofibers (743.6 ± 334.3 nm) (Figure 1B) (*p* < 0.001) in the diameter measurement. Additionally, the vildagliptin-eluting nanofibers had a smaller pore area distribution (3405 ± 1437 nm^2^) than drug-free PLGA nanofibers (8826 ± 4906 nm^2^) (*p* < 0.001). The polymer concentration in the solution for electrospinning has an influence on the surface tension and viscosity of the liquid and decides the solution into various diameters of nanofibers as the different concentration of polymers. With a higher polymeric concentration, nanofibrous membranes in the drug-free PLGA group have a higher viscosity owing to enhanced molecular chain entanglement [22,23]. The inclusion of vildagliptin reduced the PLGA concentration in the polymeric solution, leading to a lower viscosity and fewer polymer chain entanglements. The solution thus had less strength and was less able to resist the external electric force during spinning, leading to nanofibers with smaller diameters as well as greater inter-fiber pores [22,24]. 

The assessed mechanical properties in Figure 2A suggested that the vildagliptin-eluting group has comparable tensile strength with the control (drug free) group (1.81 ± 0.07 MPa for the vildagliptin group vs. 1.80 ± 0.03 MPa for the control group) (*p* = 0.701). The PLGA nanofibers without vildagliptin showed a higher elongation at breakage (233.7 ± 10.6 %) than the vildagliptin-eluting nanofibers (33.5 ± 1.8%) (*p* < 0.001). The mechanical stability of fabricated nanofibrous membrane could support regenerated tissue because the measured tensile moduli of PLGA-based nanofibers with or without vildagliptin were both comparable to that of rat skin [25]. The nanofibrous vildagliptin-eluting PLGA membranes also demonstrated good extensibility and flexibility, proving their performance in wound management that is tolerable for skin contraction during the process of wound healing. Figure 2B reveals the variations over time of the water retention of the vildagliptin group and drug-free PLGA nanofibers. Whereas the vildagliptin nanofibers group attained their highest water content of 520 ± 69% in the third hour, the drug-free PLGA nanofibrous membranes reached their peak water-retention activity (49 ± 19%) in the first hour. The amount of water stored in the vildagliptin-eluting nanofibers was higher than that in the drug-free PLGA nanofibers (*p* < 0.001) (see Appendix A for the variations in water content). The physical properties of PLGA depend upon the ratio of lactide to glycolide, and relatively glycolide-rich PLGA copolymers are hydrophilic [26,27] when drug-free PLGA is diluted by another compound. A variety of compounds associated with hydrophilicity, such as vildagliptin, that have been blended into PLGA may alter its degradation, release kinetics, and water content.

Figure 3 presents that the measured water contact angles of vildagliptin-loaded nanofibers and drug-free PLGA nanofibrous membranes are 95.2 ± 2.2° and 113.9 ± 4.9°, respectively. The mixture of vildagliptin and PLGA greatly decreased the hydrophobicity of the electrospun nanofibrous membranes (*p* = 0.004). The presence of vildagliptin, a hydrophilic compound [28], in the electrospun nanofibers thus modified their hydrophilicity and water-retention capacity within membranes. Providing and maintaining an adequate moist environment prevents the dehydration of the wound bed, facilitates wound healing, and promotes the growth of new tissue [29,30].

### 2.1. In Vitro Release Curves of Vildagliptin and EPCs Migration Assay

Transwell spreading and migration assays were performed to evaluate the ability of the cell to respond directionally to the eluents that released vildagliptin [31]. The EPCs migration assay (Figure 4A–D) revealed significantly more cell migration on day 7 (524 ± 59 cells/mm^2^) and day 28 (754 ± 56 cells/mm^2^) than that of DPBS (392 ± 36 cells/mm^2^) alone (all *p* < 0.001) as a control. EPCs play the key roles in the period of endothelial replacement and repair, as they remarkably contribute to endothelial neo-angiogenesis and homeostasis for various detrimental attacks. In patients with diabetes, the reduced EPC number is mainly due to an imbalance between the responses of endothelial repair and injury [32,33]. DPP4 inhibitors have pleiotropic effects through complex cellular mechanisms, such as decreasing mononuclear macrophages’ infiltration and increasing the number of circulating EPCs [34,35]. Figure 4E presents the daily release curve of vildagliptin in vitro. The vildagliptin-eluting membranes continuously released vildagliptin for 30 days, with an initial burst phase in the first day (156.8 µg/mL), followed by an increasing exponential linear drug release (n > 1 in Equation (1)) until a second peak at day 21 (247 µg/mL), followed by a stably decreasing linear elution (n < 1) until day 30 (>98 µg/mL). The total release amount of vildagliptin in 30 days was around 2560 µg, accounting for approximately 45% (*w*/*w*) of active substance of vildagliptin added into drug-eluting products:Concentration (µg/mL) = A × 10^n (day)^(1)
where A is a drug-dependent constant and n is the slope.

Appendix A displays the FTIR assay results for the drug-free PLGA and vildagliptin-eluting PLGA nanofibrous membranes. The FTIR spectrum of vildagliptin showed prominent peaks at 3110–3700 cm^−1^ (broad), representing OH and N–H stretching vibrations. The new vibration peaks at 2250 cm^−1^ was assigned to the nitrile stretching band [36]. In the case of the formulation (vildagliptin/PLGA), overlapping of the characteristic peaks was observed, indicating the adequate incorporation and stable nature of the drug during electrospinning of the nanofibrous membranes with the solvent of HFP.

The release characteristics of the nanofibrous membranes are influenced by the type of the polymer and the drug that forms the fibers, as electrospun nanofibers showed a significant extent of burst because of the quick release of the drug left at the fiber surface and the delayed release of the drug through the fiber bulk owing to limited water uptake [37,38]. With more drug diffusion through the inner part of the polymeric membrane in the second stage, the drugs would preferably diffuse out through the inner or near surface of nanofibrous membranes [39]. In total, the biodegradable vildagliptin-eluting PLGA membranes provided effective concentrations of vildagliptin for four weeks.

### 2.2. Diabetic Wound Healing and Histological Examination

Figure 5A–F display the change in wound healing percentage in both groups (the vildagliptin-eluting PLGA group and drug-free PLGA group as a control) on days zero, seven, and fourteen following treatment. On days seven and fourteen, the residual wound area percentage in the vildagliptin group was visibly smaller than that in the other group (day 7: 22.9 ± 9.5 vs. 59.9 ± 5.7%, *p* < 0.001; day 14: 11.4 ± 3.0 vs. 18.7 ± 2.6%, *p* < 0.001) (Figure 5G).

Diabetes usually impairs the mechanism of normal wound healing at various stages, which results in delayed wound healing and leads to peripheral vascular disease and lower limb amputations [40]. DPP-4 is an important factor and a potential factor that causes diabetic wounds to persist; therefore, DPP-4 inhibitors may accelerate healing using the opposite effect [41,42,43]. On day 7 and day 14, wounds treated with vildagliptin-eluting nanofibrous dressings exhibited faster healings than those in the control group (all *p* < 0.001).

Figure 6 presents the images captured as part of the histological investigation, Western blot, and MFI of immunostaining for CD31. In both groups, without a significant inflammatory response, the nanofibrous membranes without vildagliptin were noted to have the defect of ECM deposition below the epidermis (Arrows, Figure 6B). The immunofluorescence of CD31 expression (Figure 6C,D) (normalized to a DAPI nuclear stain) was significantly increased in the dermis in the vildagliptin group. The vildagliptin/PLGA group had higher protein expression levels of CD31 (intensity ratio: 1.73 ± 0.39 vs. 0.45 ± 0.17 *p* = 0.006) (Figure 6E,F) using Western blotting in wounds after treatment. The immunofluorescent stain of CD31/DAPI ratio exceeded that in the control group (2.2 ± 0.5 vs. 0.7 ± 0.1, *p* < 0.001) (Figure 6G). The proliferative stage of wound healing is indicated by angiogenesis, granulation tissue formation, and re-epithelialization involving endothelial cells, fibroblasts, and keratinocytes [44]. Diabetes-related alterations can impair angiogenesis, ECM deposition, and wound recovery owing to diminished cell migration and proliferation, decreased response to growth factors, and reduced cytokine secretion [45]. Neovascularization results from circulating progenitor and stem cells that are differentiated into mature endothelial cells or from the migration and proliferation of CD31-positive endothelial cells of pre-existing blood vessels [46,47]. Circulating endothelial cells participate in the formation of blood vessel during pathological and physiological processes of inflammation and wound healing [48]. The impairment of diabetic wound healing is considered by reduction in hypoxia-induced neovascularization in the skin. Treatment with a DPP4 inhibitor can enhance angiogenesis and wound healing in diabetic wounds [49]. The increase of GLP-1 level caused by DPP4 inhibition in the wound area for improving the multiple cells’ migration indicates that DPP4 inhibitors have a key role in the acceleration of wound healing process [41]. The findings indicate that the vildagliptin delivered from a drug-eluting membrane enhances wound recovery by attracting CD31 positive endothelial cells that migrate to the diabetic wound area. Future work will be required to identify the concentration of vildagliptin in plasma or in a wound to clarify the association between different drug delivery systems.

## 3. Materials and Methods

### 3.1. Fabrication of Nanofibrous PLGA Membranes

PLGA (lactide/glycolide ratio of 50:50) (Resomer RG 503, Boehringer, Germany) with a molecular weight of 24–38 kDa was used in the fabrication of membranes. Vildagliptin (C_17_H_25_N_3_O_2_) and hexafluoroisopropanol (HFP) were obtained (Sigma-Aldrich, Saint Louis, MO, USA) and mixed for the following products.

Vildagliptin-eluting nanofibrous PLGA membranes (vildagliptin, 40 mg; PLGA, 240 mg; HFP: 1000 µL) and drug-free PLGA membranes (PLGA, 280 mg; HFP: 1000 µL) were fabricated at around 25 °C using the especially designed electrospinning setup [50]. During electrospinning, the needle with an inside diameter of 420 nm had a high-voltage direct current connection (35 kV and 4.16 mA). The electrospun voltage was 35 kV, feed rate was 3.6 mL/h, and tip to collector distance was 120 mm. The electrospun products were kept in a vacuum drying oven at 40 °C for three days to evaporate HFP.

### 3.2. SEM Observation

The distributions of the diameter and pore area were calculated using SEM (Hitachi S-3000N, Tokyo, Japan) images of 100 randomly selected nanofibrous membranes of both products (*n* = 3) in Image J image software (National Institutes of Health, Bethesda, MD, USA).

### 3.3. Mechanical Properties of Nanofibrous Membranes: Tensile Strength and Elongation at Breakage

Consistent with the ASTM D638 standard, a Lloyd tensiometer (AMETEK, Berwyn, PA, USA) was used to evaluate the mechanical properties, tensile strength (MPa), and elongation at breakage (%) of nanofibrous membranes.

### 3.4. Contact Angle of Water

The water contact angles of the vildagliptin-eluting and drug-free PLGA membranes were evaluated using a video monitor and a water contact angle analyzer (First Ten Angstroms, Portsmouth, VA, USA).

### 3.5. Water-Retention Activity

The water-retention activities of both nanofibers at 0.5, 1, 2, 3, 8, and 24 h were evaluated. 

### 3.6. In Vitro Release of Vildagliptin

The release characteristics of vildagliptin-eluting PLGA nanofibrous membranes were measured from the dissolution medium (Dulbecco’s phosphate-buffered saline (DPBS)) using high-performance liquid chromatography (HPLC) (Hitachi L-2200 Multisolvent Delivery System, Tokyo, Japan). Briefly, the membranes were placed in glass test tubes with 1000 µL of DPBS. All tubes were incubated at 37 °C and the dissolution medium was collected at 24 h intervals. Fresh DPBS (1000 µL) was added at the beginning of each interval for 30 days. An Ascentis^®^ C18 (4.6 × 150 mm, 5 µm) HPLC column was conducted for the separation. The absorbance value at a 210 nm wavelength was observed with a 1000 µL/min flow rate. The mobile phase consisted of acetonitrile and 0.1 M phosphate buffer (15/85, *v*/*v*) [51]. 

### 3.7. Fourier Transform Infrared Spectroscopy 

To obtain the infrared spectrum of the vildagliptin-loaded membranes, Fourier transform infrared (FTIR) spectroscopy was performed by a spectrometer (Bruker Tensor 27, Billerica, MA, USA) with a resolution of 4 cm^−1^ in the absorption mode, leading to a total of 32 scans. The nanofibrous products (2 mg) were prepared by grinding with potassium bromide (KBr) and were then pressed into a disc, and the absorption spectrum was monitored in the range of 400–4000 cm^−1^.

### 3.8. EPCs Migration Assay 

Transwell filters (8.0 µm pores, Costar, Cambridge, MA, USA) were used for the migration assay [52,53]. EPCs were provided by the Laboratory of Molecular Pharmacology (Chang-Gung University, Taoyuan, Taiwan). At each time point, data were obtained from five randomly selected lower surface fields that had been treated with the eluent from vildagliptin-eluting membranes.

### 3.9. Test of Diabetic Wound Healing In Vivo

Twenty Sprague–Dawley streptozotocin (STZ) (Sigma, Burlington, MA, USA)-induced diabetic rats were used in this investigation. The animals were well cared for and their use was supervised by a licensed veterinarian following institutional approval (Chang Gung University CGU No. 14-045, Taoyuan, Taiwan). With repeated verification, blood glucose levels of over 300 mg/dL in the rats were measured seven days after the administration of STZ. During the whole duration of the experiments, all animals were housed in individual cages in a central animal care facility with a 12 h light/dark cycle and given free access to standard rodent chow and water, and the temperature and humidity were controlled.

The STZ-induced diabetic rats were separated into two groups, each of ten rats, for treatment with drug-eluting membranes (Vildagliptin/PLGA) or drug-free PLGA nanofibrous membranes. After two weeks, the rats were sacrificed using a carbon dioxide chamber.

### 3.10. Immunofluorescence and Western Blotting

To perform immunostaining, primary antibodies against CD31 (1/1000, ab124432, Abcam) were then stained with secondary antibodies with AF 546 goat anti-mouse secondary antibodies (1/500, Life Technologies) for 8 h at 4 °C. The nuclear stain of 4,6-diamidino-2-phenylindole (DAPI) (1/2000 dilution in PBST, 120 min) was used the following day. The mean fluorescence intensity (MFI) was determined as the mean over regions of interest using ImageJ software. The target protein was normalized to that of MFI of DAPI as an internal control for sample-to-sample variations. For Western blotting, samples were incubated with primary antibodies (CD31). The membranes were incubated and rinsed with secondary antibodies for one hour with anti-rabbit, IgG secondary antibodies. Densitometric analysis of protein expression was normalized to the expression of glyceraldehyde-3-phosphate dehydrogenase (GAPDH) (1/10,000, ab8245, Abcam) using ImageJ software. The statistical results were calculated using a minimum of three experimental runs.

### 3.11. Statistics and Analysis of Data 

Data are shown as the mean ± standard deviation. The means of continuous variables and normally distributed data were evaluated using the unpaired Student’s t-test; otherwise, the Mann–Whitney U test was performed. Differences were regarded as statistically significant at *p* < 0.05, as determined by SPSS software (version 17.0 for Windows; SPSS Inc., Chicago, IL, USA).

## 4. Conclusions

Nanofibrous biodegradable drug-eluting PLGA membranes that sustainably released vildagliptin to heal diabetic wounds were fabricated. An effective drug concentration was provided from the nanofibrous vildagliptin-eluting membranes for 30 days. Vildagliptin-eluting PLGA nanofibrous membranes had smaller fiber diameters and pore area, as well as lower hydrophobicity and water-retention capacity, than drug-free PLGA nanofibers. The fabricated vildagliptin-eluting PLGA nanofibers also enhanced the diabetic wound recovery and the level of angiogenesis. These results demonstrate that nanofibrous drug-eluting PLGA membranes are effective in terms of providing sustained drug release and accelerating the diabetic wound healing.

## Figures and Tables

**Figure 1 pharmaceuticals-15-01358-f001:**
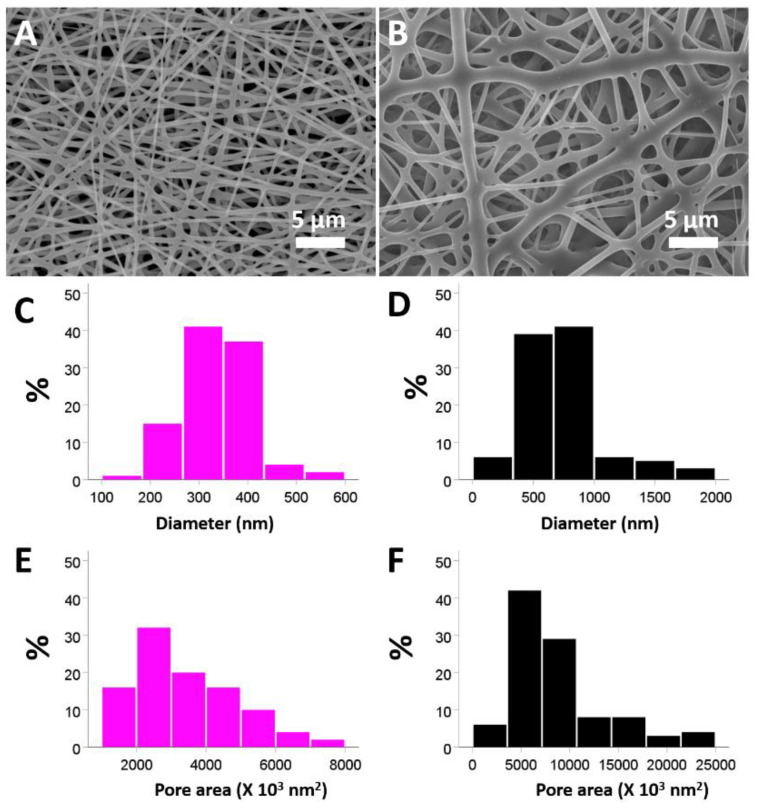
SEM images of electrospun nanofibers composed of PLGA with (**A**) and without (**B**) vildagliptin. Properties of vildagliptin-loaded PLGA (**C**) and drug-free PLGA (**D**) were calculated for the diameters. The pore space in both groups was assessed using Image J software for the vildagliptin group (**E**) and drug-free PLGA groups (**F**) (scale bar: 5 μm).

**Figure 2 pharmaceuticals-15-01358-f002:**
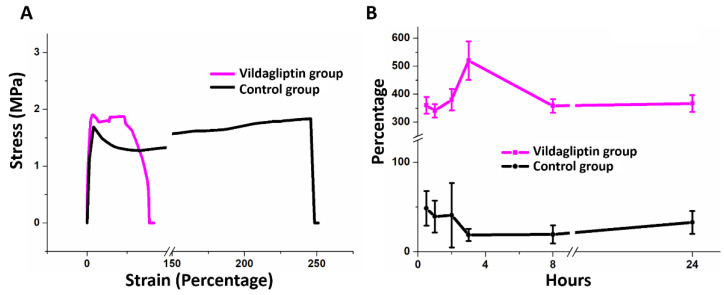
Stress–strain curves of nanofibrous membranes of the vildagliptin and control (drug-free) groups (**A**). Black corresponds to drug-free PLGA and red corresponds to vildagliptin/PLGA nanofibrous membranes. Variation in the water-retention capacity of nanofibers for 24 h (**B**).

**Figure 3 pharmaceuticals-15-01358-f003:**
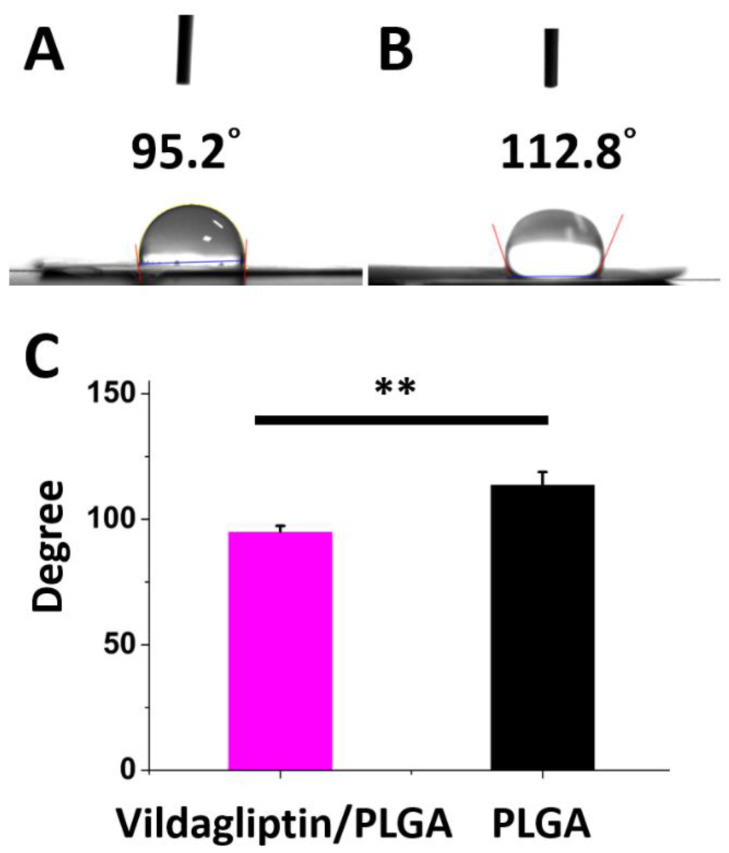
The photos (**A**) and measured water contact angles (**B**) of biodegradable nanofibrous membranes both with and without vildagliptin (**C**) (** *p* < 0.01).

**Figure 4 pharmaceuticals-15-01358-f004:**
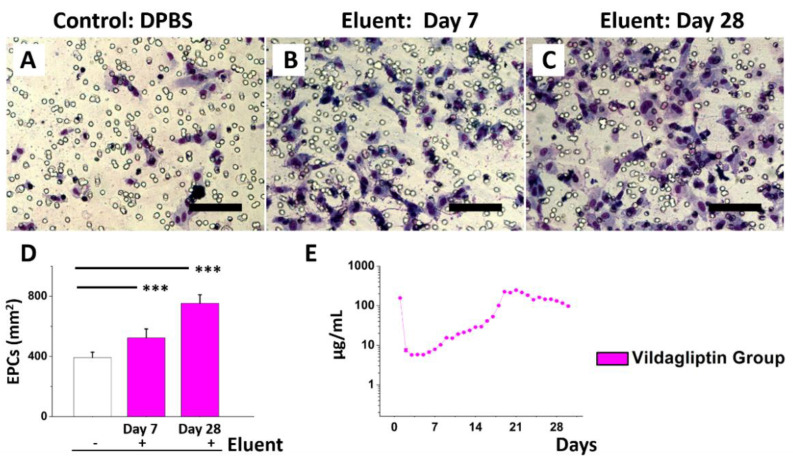
Eluent of vildagliptin/PLGA membranes for EPCs migration test. EPCs added with DPBS only (**A**), day 7 eluent (**B**), or day 28 eluent (**C**). Eluents with vildagliptin increase EPCs’ migration (**E**). In vitro release of vildagliptin (**D**) (*** *p* < 0.01).

**Figure 5 pharmaceuticals-15-01358-f005:**
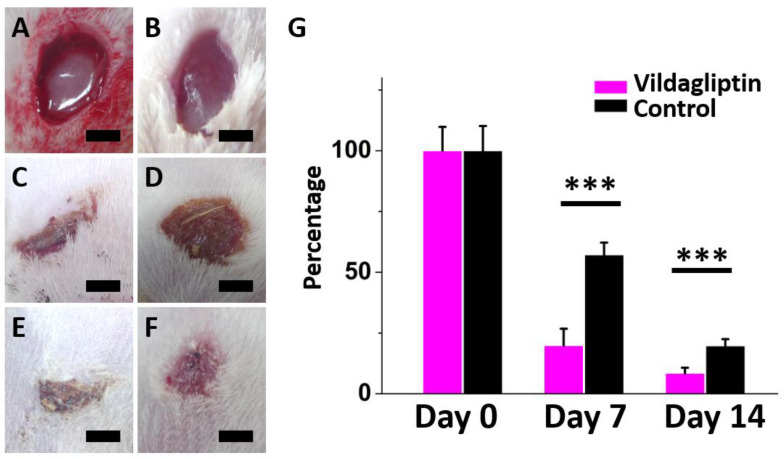
Progress of wound recovery following treatment in both groups on day 0 ((**A**): vildagliptin group; (**B**): control group)), day 7 ((**C**): vildagliptin group; (**D**): control group)), and day 14 ((**E**): vildagliptin group; (**F**): control group)). The residual percentage of diabetic wounds treated with both groups after one and two weeks (**G**) (*** *p* < 0.001) (scale bar = 5 mm).

**Figure 6 pharmaceuticals-15-01358-f006:**
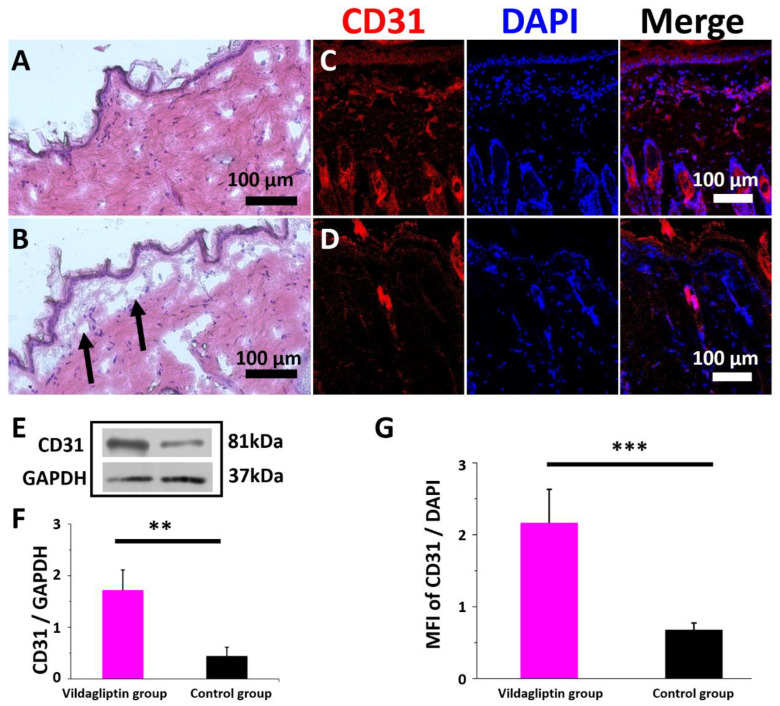
Histological images of the vildagliptin-eluting (**A**) and drug-free PLGA (**B**) group on day 14. Immunofluorescence of CD31 in the vildagliptin/PLGA (**C**) and drug-free PLGA (**D**) groups. Intensity ratio is obtained by densitometry as the ratio of the target protein to GAPDH density (**E**). Western blot and immunocytochemical quantification for CD31 relative to GAPDH or DAPI using the CD31/GAPDH ratio (**F**) or mean fluorescence intensity (MFI) of CD31/DAPI (**G**), respectively (scale bar = 100 μm) (** *p* < 0.01, *** *p* < 0.001).

## Data Availability

Data is contained within the article or Appendix A.

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
