# Peer review of "Nanofibrous Vildagliptin/PLGA Membranes Accelerate Diabetic Wound Healing by Angiogenesis"

_pharmaceuticals, 2022, doi:10.3390/ph15111358_

Round 1

Reviewer 1 Report

Authors presented valuable study concerning nanofibrous drug-eluting poly-D-L-lactide-glycolide (PLGA) scaffolds that sustainably release a high concentration of vildagliptin. The aim of developing such scaffolds was to accelerate wound healing in diabetes. The idea of the study is interesting and the reason of conducting this research is justified. The problem of diabetic wounds is very high worldwide so there is constant need for developing new therapies- scaffold with the function of local release would be much beneficial.

Manuscript is well written and is divided into logical part. however there are some misunderstanding in the text that should be explained or correct.

·         2.6. In-vitro release of vildagliptin

Please provide details concerning the course of experiment such as: the type and amount of the medium, the time points, how often the medium was replaced

·         „The vildagliptin-eluting PLGA scaffolds (336.0 ± 69.1 nm) (Fig.

1A) were lower than those of the virgin PLGA scaffolds (743.6 ± 334.3 nm)”please explain. Is it concerning the size? It should be rather smaller

·         There is some misunderstanding in the sentence:

Figure 5. Progress of wound healing after treatment in virgin PLGA group on day 0 (A), day 7 (B), and day 14 (C). The residual percentage of diabetic wounds treated with vildagliptin/PLGA group after one and two weeks (G). (*** p < 0.001). (Scale bar = 5 mm).

What is D,E,F? It is not explained in the text.

Reviewer 2 Report

I have been asked to review a paper entitled ‘Nanofibrous Vildagliptin/PLGA Membranes Accelerates Diabetic Wound Healing by Angiogenesis’. Herein are some selective comments towards the authors that might improve this work.

Abstract

It is important to improve the English language in the Abstract and the manuscript as a whole.

Introduction

Font size changed during reading, please make the size consistent throughout the manuscript.

Authors should add more on the benefits of using nanofiber scaffolds in tissue engineering and wound healing.

Also, add more on the benefit of electrospinning and a brief description of this technology.

Material and methods

Add the electrospinning parameters

Change virgin PLGA scaffolds to one of these synonymous (blank, placebo or drug-free) and change it throughout the manuscript.

Generally, a more detailed description of all methods should be considered

How the animals were maintained and sacrificed at the end of the study

Results and Discussion

Better to use water retention instead of water storage throughout the manuscript

Authors should be consistent, either using Figure or its abbreviation Fig.

All Figures' legends have mistakes. Need proper checking.

In Figure 5, since letters D-F are not described, I’m assuming they represent vil-loaded fibers. Why the size/depth of the wound differs between A and D. This difference may actually affect the healing duration, especially that on day 14, the wound residual % was very close to the drug-loaded fibers.

Why there isn’t an untreated group of animals and commercially available treatment products as experimental controls? There were more animals needed in a single group (10 rats in one group).

Conclusion

Needs to be improved.

This work is very close to the authors' previous work doi: 10.2147/IJN.S211898

I don't see any novelty and am concerned about animal treatment, as there was no description of how the authors maintained and sacrificed the animals at the end of the study. 

The authors also used 10 animals per group and didn't evaluate an untreated or commercially available product. Using 10 animals in the group is more than what was needed to prove the efficacy of the fibers. 

Reviewer 3 Report

The paper "Nanofibrous vildagliptin/PLGA membranes accelerates diabetic wound healing by angiogenesis" (Chen-Hung Lee et al.) described the synthesis and characterization of nanofibrous biodegradable vildagliptin-eluting PLGA membranes. The authors used SEM to evaluate the diameters and pore areas of electrospun nanofibers and to test the efficacy of drug release in vitro. In addition, immunohistochemistry was used to investigate the effects of nanofibrous vildagliptin-eluting PLGA scaffolds on diabetic wound repair.

The paper is comprehensive and might be worth considering for publication in Pharmaceuticals; however, the authors must significantly revise and improve it first.

Comments and Suggestions for Authors:

1. The authors must include a current review of literature on obtaining and characterizing similar materials. Please also indicate the innovative ideas in the described work.

2. I'm missing some chemical analysis of developed materials, such as FTIR spectroscopy. Of course, the FR-IR spectra of the drug and the PLGA matrix should be tested concurrently.

3. The authors conclude that "the presence of vildagliptin, a hydrophilic compound [27], in the electrospun nanofibers thus modified their hydrophilicity and water storage capacity within scaffolds." That is correct, but my question is: how much active substance by weight (or mole) was added to the virgin PLGA to create the final material?

4. What is the patient's daily vildagliptin dosage? Can this dose be related to the amount of drug released from the authors' materials?

5. Figure 4E - Please add a unit to the X axis.

6. The authors' claim on page 6 that...... "followed by a stable elution until day 30 (> 98 g/ml)" is overstated. The authors observe a significant drug release (burst release) on the first day, after which the amount of drug released decreases, then increases to day 21, and then decreases again in the following days. So, where is this stability in the release profile? In general, the authors should continue the drug release experiment until they obtain a stable profile. Statistics, on the other hand, should be included.

7. The authors should also use appropriate mathematical models to demonstrate how the active substance is released from the developed material (drug release mechanism).

8. In addition, please discuss and compare the results of in vitro tests with data from the literature.

Round 2

Reviewer 2 Report

Thank you for addressing all my points. 

Reviewer 3 Report

Dear Authors,

In my opinion, the manuscript has been sufficiently revised to be published in Pharmaceuticals. My only objections are minor grammatical errors.